# Changes in Sleep Satisfaction of Korean Adolescents in the Pre- and Post-COVID-19 Eras and Its Effects on Health Behaviors

**DOI:** 10.3390/ijerph20031702

**Published:** 2023-01-17

**Authors:** Dahyeon Lee, Kang-Sook Lee, Sejin Kim, Woohyun Chung, Jeung Jegal, Hyeonjung Han

**Affiliations:** 1Department of Health Promotion, Graduate School of Public Health and Healthcare Management, The Catholic University of Korea, 222 Banpo-daero, Seocho-gu, Seoul 06591, Republic of Korea; 2Department of Preventive Medicine, College of Medicine, The Catholic University of Korea, 222 Banpo-daero, Seocho-gu, Seoul 06591, Republic of Korea; 3Department of Occupational and Environmental Health, Graduate School of Public Health and Healthcare Management, The Catholic University of Korea, 222 Banpo-daero, Seocho-gu, Seoul 06591, Republic of Korea

**Keywords:** adolescents, sleep quality, physical activity, health behaviors, internet use, smartphone use, COVID-19

## Abstract

Adolescence is a crucial period for cognitive and psychological development and physical maturation. During this period, hormonally influenced circadian rhythms lead to reduced hours of sleep, and it is important to determine whether sleep quality is sufficient for fatigue relief. Non-face-to-face classes during coronavirus disease-2019 (COVID-19) potentially affected adolescents’ sleep quality, psychological state, amount of physical activity, smoking, alcohol consumption, and internet (smartphone) use. We investigated the effects of the COVID-19 situation on adolescents’ sleep satisfaction and its relation to the aforementioned factors. Data of 109,281 adolescents collected via an online survey, conducted from 3 June 2019 to 12 July 2019 and from 3 August 2020 to 13 November 2020, were analyzed. Health status comparison between the satisfactory and unsatisfactory sleep groups yielded significant results (odds ratio [OR] = 1.10, confidence interval [CI] = 1.04–1.17) for 2020. In both groups, perceived health was worse in 2019 than in 2020 (OR = 2.72, CI = 2.53–2.92). During COVID-19, non-face-to-face classes increased adolescents’ sleep satisfaction. Their psychological state improved, while amount of physical activity (muscle-strengthening exercises), average weight, and internet (smartphone) use increased. Smoking and alcohol consumption decreased.

## 1. Introduction

Sleep is essential for human beings to recover from physical and mental fatigue and to resume regular physical activities [1]. Adolescence is an important period for cognitive and psychological development and physical maturity; it is essential to ensure sufficient sleep for a healthy lifestyle. In addition, circadian rhythm changes are hormonally influenced during this period, leading to decreased hours of sleep. It is critical to determine whether sleep quality, rather than the length of sleep, is sufficient to recover from fatigue [2]. 

Lack of sleep during adolescence can negatively affect mental health [3]. Insufficient sleep leads to increased depression and emotional instability [4] and affects adolescents’ mental health and lifestyle, thereby significantly impacting their overall quality of life [5]. Korean adolescents have insufficient sleep compared with adolescents in other countries because they go to school early and return home late [6]. 

A survey on subjective happiness conducted among adolescents in major Organisation for Economic Co-operation and Development (OECD) countries reported that Korean adolescents had low happiness levels and were under high stress due to excessive competition for university entrance exams [7]. OECD is an international organisation that works to build better policies for better lives. 

Moreover, the subjective happiness of adolescents is related to their amount of physical activity. Increased participation in physical activity and sports classes correspondingly increases the level of subjective happiness [8,9,10]. An increase in the number of days of physical activity and muscle exercise resulted in higher levels of subjective happiness and improved mental and perceived health status [11]. The time spent using devices such as smartphones reduces the amount of physical activity, which can lead to obesity in childhood and adolescence [12]. 

Korean adolescents often use smartphones extensively in the early hours of the morning [13], and internet (smartphone) addiction causes poor sleep quality and depression [14]. Individuals with high stress and depression may lack mood management techniques, rendering them more vulnerable to internet addiction [15], resulting in a vicious cycle that deteriorates sleep quality. The stress caused by the university entrance exams and increased internet usage time affects sleep quality [16,17]. Smoking and alcohol consumption among adolescents cause chronic sleep problems [18]. Additionally, smoking is associated with anxiety-induced sleep disturbances [19]. Thus, various factors can affect adolescents’ sleep and it is difficult to determine the exact causes by considering only one factor [20]. 

These health-related behaviors are interconnected with complex interactions [21,22]. Understanding their individual and combined influences and simultaneously analyzing them is vital for health promotion programs [21,22]. 

COVID-19 causes headaches caused by acute sinusitis along with respiratory diseases [23]. COVID-19 and vaccines can cause immune response symptoms [24]. Due to the COVID-19 situation, schools in Korea conducted non-face-to-face classes from March 2020 to October 2021. 

Adolescence is a period of turmoil, and it is vulnerable to such external changes because of instability and psychological imbalance [25]. In particular, it affects the physical activity and psychological health of adolescents [10]. The COVID-19 pandemic period increased screen time and affected eating habits, physical activity, and sleep quality [25]. Previous studies conducted in other countries reported that students were greatly affected by the COVID-19 situation and developed symptoms of depression, and countermeasures, such as promoting good sleep and resolving concerns about COVID-19, were suggested [26]. Moreover, studies reported changes in sleep patterns and phase delays in children and adolescents with COVID-19 [27]. The non-face-to-face educational environment during COVID-19 affected adolescents’ sleep satisfaction, psychology, physical activity, smoking, drinking, and internet (smartphone) use. It is therefore important to compare sleep conditions before and after COVID-19 and analyze health behavior. 

Several studies reported on the importance of adequate sleep quantity and sleep quality [4,6]. However, only a few focused on post-sleep fatigue relief. Based on an earlier study, which reported that sleep loss was compensated by increasing the total hours of sleep in case of low sleep efficiency [17], this study aimed to analyze the sleep satisfaction and to investigate the effect of the COVID-19 situation on adolescents’ sleep satisfaction and whether such effects were related to adolescents’ psychological state, physical activity, smoking, alcohol consumption, and internet (smartphone) use.

## 2. Materials and Methods

### 2.1. Research Participants and Data Collection

The Youth Health Behavior Survey is an anonymous self-entry online survey conducted on students aged 13 to 18 to understand the health behavior of Korean adolescents. The Korea Centers for Disease Control and Prevention under the Ministry of Health and Welfare carried it out every year since 2005. This study used data from online surveys of youth health behavior in 2019 and 2020. This study was based on data collected from an online survey on adolescents’ health behavior that was conducted from 3 June 2019 to 12 July 2019 and from 3 August 2020 to 13 November 2020. A total of 112,251 responses were considered. After cleaning missing data, a total of 109,281 (55,747 in 2019 and 53,534 in 2020) responses were used for the final analyses. This study was approved by the Institutional Review Board at the Catholic University of Korea (MC22ZASI0110).

### 2.2. Methods

#### 2.2.1. Demographic Variables

Gender was classified into “boys” and “girls”; size of the city as “Rural area (Gun)”, “metropolitan city”, and “small–medium city”; schools where the participants were currently attending as “middle school”, “general high school”, “specialized high school”, and “no response”; school achievement and economic status as “upper”, ”middle-upper”, “middle”, “middle-low”, and “low” on a 5-point Likert scale. These items are combined as “upper”, “middle”, and “low” in the tables. Economic changes due to family circumstances and the COVID-19 pandemic in the past 12 months were collected for the year 2020 only, and only these data were analyzed. The types of residence were classified into the five following groups for analyses: “living with family”, “living in relatives’ house”, “living in a boarding house or living alone”, “living in a dormitory”, and “living in a childcare facility”. Mean analysis was performed for age, height, weight, and BMI. 

#### 2.2.2. Mental State, Physical Activity, and Relief from Fatigue after Sleep 

The following data were selected from the survey results to examine the changes in the psychological state and amount of physical activity in adolescents in the pre- and post-COVID-19 eras. Regarding the psychological state, “perceived health” and “awareness of everyday stress” were examined and each question comprised a 5-point Likert scale. Thereafter, the items with fewer answers were combined and presented in the tables. “Perceived health” was classified into “healthy”, “neutral”, and “unhealthy”. “Awareness of everyday stress” was classified into “very much”, “much”, “moderate”, and “mild”. A question on the experiences of depression in the last 12 months was answered as “yes” or “no”. The amount of physical activity was measured as “the number of days of high-intensity physical activity in the past seven days”, “the number of days of muscle-strengthening exercise in the past seven days”, and “weight control efforts in the past 30 days”. To examine if the adolescents’ sleep quality was satisfactory, the participants were queried whether the sleep in the last seven days was sufficient to recover from fatigue. The sleep satisfaction was measured using a 5-point Likert scale as follows: “satisfied”, “slightly satisfied”, “neutral”, “slightly dissatisfied”, and “dissatisfied”. 

#### 2.2.3. Experience with Smoking, Alcohol Consumption, and Internet (Smartphone) Use

To examine the lifestyle changes of adolescents in the pre- and post-COVID-19 eras, the following data were selected from the survey results: Regarding smoking, questions on “any experience of regular smoking cigarettes”, “any experience of using liquid e-cigarettes”, and “any experience of using e-cigarettes” were answered as “yes” or “no”. To examine the experiences of exposure to secondhand smoke, the participants were queried on the number of days that they were exposed to the following: “secondhand smoke at home”, “secondhand smoke (indoors) at school”, and “secondhand smoke (indoors) in public places”. To assess alcohol consumption experience, questions on any alcohol consumption experience were answered as “yes” or “no”. Ease of purchasing cigarettes and alcohol was examined, and each question comprised a 5-point Likert scale. These items are combined and are indicated in the table as “no attempt to purchase”, “impossible to purchase”, and “easy to purchase”. To assess whether the internet (smartphone) was overused, related questions were answered with a “yes” or “no” response. This question queried whether the internet was used in 2019 and smartphones in 2020. 

### 2.3. Data Analysis

The variables of this study were analyzed using the IBM SPSS Statistics 28.0.1 program (IBM Co., Armonk, NY, USA). Demographic and sociological information on the adolescents was collected through the survey, and the participants’ general characteristics were analyzed using cross-analysis. For comparison of health status according to the satisfactory sleep among adolescents in the pre- and post- COVID-19 eras, participants who responded with ”satisfactory“, “slightly satisfied“, and “neutral” for post-sleep fatigue were combined into a group that had a “satisfactory sleep”, which was used as a reference group. Participants who responded with “slightly dissatisfied” and “dissatisfied” for sleep satisfaction were combined into a group with “unsatisfactory sleep”. Binary logistic regression analysis was performed by adjusting age and sex. 

Statistical significance was determined based on the significance level of 5%. 

## 3. Results

### 3.1. General Characteristics of Research Participants

Among the 109,281 participants in this study, 56,746 (51.9%) were boys; 53,022 (48.5%) were living in small–medium-sized cities; 56,966 (52.1%) were attending middle schools; 33,089 (30.3%) were “middle” in their school achievement; and 52,594 (48.1%) were “middle” by economic status. The most common type of residence was “living with family” (103,947 [95.1%]). Regarding the comparison of the changes in economic status before and after the COVID-19 pandemic, 5322 students (9.9%) reported economic changes due to family circumstances, whereas most reported not having economic changes (*n* = 21,350, 39.9%). The average age, height, weight, and BMI were 15 years, 165.8 cm, 59.3 kg, and 21.4 kg/m^2^, respectively (Table 1).

### 3.2. Mental State and Physical Activities by Year (from 2019 to 2020) 

The number of participants who perceived their health status as “healthy” and “not healthy” decreased from 24,333 (43.6%) to 22,848 (42.7%) and increased from 3698 (6.6%) to 3689 (6.9%), respectively. The rate of feeling everyday stress as “very much” and “mild” decreased from 6410 (11.5%) to 4379 (8.2%) and increased from 8708 (15.6%) to 9697 (18.1%), respectively. The number of adolescents who did not experience depression in the past 12 months increased from 40,264 (72.2%) to 40,138 (75%). Participants with 0 days of high-intensity physical activity in the last 7 days increased from 17,517 (31.4%) to 20,394 (38.1%). Participants who performed more than five days of muscle-strengthening exercise in the past seven days increased from 5640 (10.1%) to 6639 (12.4%). Participants who did not make an effort to control their weight in the last 30 days decreased from 26,466 (47.5%) to 24,274 (45.3%). Participants who were “slightly satisfied” and “dissatisfied” with sleep hours in the last seven days increased from 8540 (15.3%) to 10,983 (20.5%) and decreased from 8982 (16.1%) to 5764 (10.8%), respectively (Table 2).

### 3.3. Experience of Smoking, Alcohol Consumption, and Internet (Smartphone) Use by Year (from 2019 to 2020) 

The number of participants who were regularly smoking cigarettes decreased from 6753 (12.1%) to 5430 (10.1%); moreover, the number of participants who were using liquid e-cigarettes and e-cigarettes decreased. Regarding the ease of purchasing cigarettes in the last 30 days, the number of participants who did not attempt to purchase cigarettes increased from 93.5% to 95.4%. The number of participants who were not exposed to secondhand smoke at home in the past seven days increased from 38,007 (68.2%) to 39,819 (74.4%). The number of participants who were not exposed to secondhand smoke at schools and in public places increased. The number of students who never consumed alcohol increased from 34,234 (61.4%) to 35,744 (66.8%). Regarding the ease of purchasing alcoholic beverages in the past 30 days, the number of participants who did not attempt to purchase alcoholic beverages increased from 91.5% to 94.1%. The use of internet (smartphone) on weekdays and weekends increased from 42,040 (75.4%) to 51,678 (96.5 and from 43,817 (78.6%) to 51,850 (96.9%), respectively (Table 3).

### 3.4. Relationship between Sleep Satisfaction and Health Status 

Regarding sleep satisfaction, the health status was compared between the satisfactory and unsatisfactory sleep groups. Economic change was investigated only in 2020, indicating significant results (odds ratio [OR] = 1.10, confidence interval [CI] = 1.04–1.17). Regarding perceived health, sleep worsened more in 2019 than in 2020 in the unsatisfactory sleep group (OR = 2.72, CI = 2.53–2.92); the satisfactory sleep group was used as reference. Depression experience was greater in 2020 (OR = 2.11, CI = 2.02–2.20). High-intensity exercises were more frequently implemented in 2020 (OR = 0.90, CI = 0.87–0.94). Muscle-strengthening exercises were performed more frequently in 2020 (OR = 0.89, CI = 0.86–0.93). Alcohol consumption experience decreased in 2020 compared to 2019 (OR = 1.45, CI = 1.40–1.51). Smoking experience increased (OR = 1.49, CI = 1.40–1.58). Exposure to secondhand smoke at home and schools increased; however, exposure to secondhand smoke in public places decreased. Internet use decreased in 2020 (Table 4).

## 4. Discussion

This study aimed to investigate the changes in psychological state, amount of physical activity, and sleep satisfaction of adolescents in the pre- and post-COVID-19 eras, and to verify the relationship between psychological state, amount of physical activity, and smoking, alcohol consumption, and internet (smartphone) use and the degree of sleep satisfaction. 

Compared with the pre- COVID-19 era, a decreased and an increased number of adolescents perceived themselves as “healthy” and “unhealthy,” respectively, in the post-COVID era. Conversely, the rate of everyday stress decreased, and the number of adolescents who did not experience depression in the past 12 months increased. An increased number of adolescents in both groups reported “satisfied” and “slightly satisfied” sleep satisfaction than in the previous year, indicating sufficient hours of sleep. Non-face-to-face classes during the COVID-19 pandemic and the more flexible waking times possibly improved sleep satisfaction. Moreover, actual hours of sleep increased. The boys’ sleep satisfaction, in particular, was more affected by wake-up time [28]. Considering that the degree of post-sleep fatigue recovery increased compared with the pre-COVID-19 era, satisfactory sleep potentially improved adolescents’ mental health [3].

Regarding the effect of sleep satisfaction on psychological state, the group with unsatisfactory sleep was more likely to experience depression in the last 12 months. These findings are consistent with those reported in previous studies that excessive or insufficient sleep and stress negatively affect mental and physical health [20]. Another study reported that increased hours of sleep were related to decreased perceived stress and suicide rates [29]. 

Regarding the effect of sleep satisfaction on the amount of physical activity, an increase in the number of participants who did not perform high-intensity physical activity and who performed muscle-strengthening exercises for more than five days a week in the past seven days was observed. This may be due to the increase in the internet (smartphone) dependence during the COVID-19 pandemic leading to decreased participation in high-intensity physical activity [30]. An earlier study reported that the level of physical activity decreased due to social distancing [31]. Conversely, it is speculated that muscle-strengthening exercise increased because it can be performed at home. Physical activity programs can be jointly developed and implemented by schools and students’ families, as was suggested in previous studies [32]. The number of participants who did not make any efforts to control their weight in the last 30 days decreased, whereas the number of those who made an effort increased. Owing to the COVID-19 pandemic, the average weight increased by 1 kg compared to the previous year, and the number of days of high-intensity physical activity decreased. This indicates the necessity of efforts to decrease body weight. 

The satisfactory sleep group had a higher rate of high-intensity physical activity and muscle-strengthening exercise in the past seven days. This result was consistent with that reported in previous studies [33]. A study from India reported that adolescents spent more time on smartphones and had less physical activity during the COVID-19 pandemic [25]. Health risk behaviors, such as a lack of physical activity and sleep, occur simultaneously, and consequently, and stress management becomes difficult [20]. Therefore, preventing health risk behaviors and ensuring sufficient sleep is vital. 

Smoking was more frequent in the unsatisfactory sleep group than in the satisfactory sleep group and this decreased in the post-COVID-19 era. Exposure to secondhand smoke deceased at home, schools, and in public places, and exposure to secondhand smoke at home and schools increased in the unsatisfactory sleep group. Adolescents value the reputation of their peers; hence, they are prone to smoking or consuming alcohol to gain recognition from their peers [34]. This practice is prominent in the unsatisfactory sleep group. It can be inferred that this group may have risky health behaviors. Smokers are twice as likely to have anxiety-induced sleep disturbances than non-smokers [19]. Adolescent smoking cessation campaigns may benefit from a network approach considering their peer influence [35]. Previous studies suggested that social factors, such as community intervention, can reduce the likelihood of teenage smoking [36].

Alcohol consumption decreased in 2020 compared with 2019. Moreover, the frequency and amount of alcohol use reportedly increased when peer delinquency increased [37]. During the pandemic, the frequency of socializing with peers decreased, resulting in a decreased alcohol consumption. Previous studies reported that many adolescents smoked and consumed alcohol with their parents during the pandemic [34]. Although alcohol consumption decreased among Korean adolescents, engaging in such habits with parents is more dangerous because of the relatively easy access to alcohol. Young adults aged 18–25 years with nocturnal sleep patterns, which included staying active till late and with poor sleep satisfaction, reportedly consumed more alcohol and tobacco and had high anxiety and impulsivity levels [26]. In individuals with a late sleep cycle the cravings for alcohol peak later. Adolescents with a poor sleep cycle and sleep satisfaction are more vulnerable to alcohol addiction [38].

Adolescents who smoke and consume alcohol have chronic sleep problems and their sleep satisfaction is poor [18]. Moreover, adolescents who begin smoking and consuming alcohol at an early age are vulnerable to suicidal thoughts and behaviors [20]. Previous studies indicated that it is important to understand the relationship with family and colleagues because peer factors are essential in preventing adolescents from smoking and consuming alcohol [39]. 

Internet (smartphone) usage increased in the post-COVID era compared with the pre-COVID-19 era. Excessive smartphone use causes psychiatric and cognitive health problems and changes in the brain; therefore, educationalists and school authorities should restrict its usage at schools [40]. In addition, a longer time spent on smartphones can lead to reduced physical activity time [12]. Previous studies reported that excessive smartphone use among adolescents was related to sleep deprivation [41]. It was reported that students’ sleep satisfaction and internet (smartphone) addiction can affect their depressive symptoms. The assessment of this addiction and sleep satisfaction should have precedence [14]. Another study reports limiting screen time in the evening can improve sleep satisfaction and daytime functions in adolescents [42]. People who limit their screen time reportedly sleep 20 min longer. Therefore, limiting smartphone use is more conducive to improving sleep satisfaction [43]. In Brazil, a study reporting on school programs that provided sleep hygiene education via health messages were delivered to parents, and adolescents indicated that these measures helped improve sleep irregularities [43]. 

Sleep affects mental health and educating adolescents on the importance of sleep is necessary. Since there is a high probability that smoking, alcohol consumption, and internet (smartphone) addiction decreases sleep satisfaction [30], adolescents who engage in these health risk behaviors should be targeted for sleep hygiene education. 

People who sleep well are more likely to wake up swiftly in the morning and have a regular breakfast [44]. In addition, the likelihood of consuming convenience foods is 1.37 times lower if a person sufficiently recovered from fatigue by sleeping [45]. Conversely, if a person consumes fewer sugary drinks, the virtuous cycle may be repeated because of more sleep [46]. These individuals are more likely to adopt healthy lifestyles, maintaining or gaining weight naturally. 

Sleep is the first part of a healthy lifestyle. It is necessary to educate caregivers and adolescents on the importance of sleep. Globally, the prevalence of sleep disorders during the COVID-19 pandemic was high, and 40% of the general public and medical personnel were affected. Sleep disturbance was higher in patients with symptomatic COVID-19 [47]. This may be because an increased proportion of indoor life and a limited amount of sunlight exposure affects the melatonin secretion cycle [48]. Nevertheless, it was reported that sleep satisfaction improved among students [49]. Since this degree of fatigue relief was higher in the post-COVID-19 than in the pre-COVID-19 era, this period of non-face-to-face classes may be an opportune time to educate students on the importance of sleep and a healthy lifestyle, with the assistance of their families. 

The limitation of this study is that the first self-report survey can lead to recall errors. The second cross-sectional study is that there is a limit to explaining the causal relationship between health behavior and sleep. Lastly, since it is a study conducted with data already collected, it is difficult to know the impact on individual sleep. In addition, as a survey conducted by a national institution, the questionnaire was written using the internet in 2019 and the questionnaire was written using a smartphone in 2020.

## 5. Conclusions

During the COVID-19 pandemic, non-face-to-face classes were conducted, and adolescents’ degree of post-sleep fatigue relief and sleep satisfaction improved. Their psychological state improved, the amount of physical activity increased mainly for muscle-strengthening exercises, and the average body weight increased. Smoking, secondhand smoke exposure, and alcohol consumption decreased, while internet (smartphone) usage increased. Since there may be environmental changes, such as the COVID-19 situation, in the future, schools or local organizations should prepare for such situations.

## Figures and Tables

**Table 1 ijerph-20-01702-t001:** Participants’ general characteristics.

		2019 N = 55,747 n (%)	2020 N = 53,534 n (%)	Total N = 109,281 n (%)	*p*-Value
Sex	Boys	29,059 (52.1)	27,687 (51.7)	56,746 (51.9)	0.177
	Girls	26,688 (47.9)	25,847 (48.3)	52,535 (48.1)	
City size	Rural area	4352 (7.8)	4192 (7.8)	8544 (7.8)	0.000
	Metropolitan	24,657 (44.2)	23,058 (43.1)	47,715 (43.7)	
	Small–medium city	26,738 (48)	26,284 (49.1)	53,022 (48.5)	
School type	Middle school	28,674 (51.4)	28,292 (52.8)	56,966 (52.1)	0.000
	General high school	21,613 (38.8)	20,099 (37.5)	41,712 (38.2)	
	Specialized high school	5021 (9)	4759 (8.9)	9780 (8.9)	
	No response	439 (0.8)	384 (0.7)	823 (0.8)	
School achievement	Upper	21,389 (38.3)	19,767 (36.9)	41,156 (37.7)	0.000
	Middle	16,848 (30.2)	16,241 (30.3)	33,089 (30.3)	
	Low	17,510 (31.4)	17,526 (32.8)	35,036 (32)	
Economic status	Upper	21,897 (39.2)	20,895 (39.1)	42,792 (39.1)	0.516
	Middle	26,856 (48.2)	25,738 (48.1)	52,594 (48.1)	
	Low	6994 (12.5)	6901 (12.9)	13,895 (12.8)	
Economic changes due to family circumstances in the past 12 months	No		48,212 (90.1)	48,212 (90.1)	
	Yes		5322 (9.9)	5322 (9.9)	
Economic changes due to the COVID-19 pandemic in the past 12 months	Very difficult		3072 (5.7)	3072 (5.7)	
	Difficult		13,189 (24.6)	13,189 (24.6)	
	No change		21,350 (39.9)	21,350 (39.9)	
	No change at all		15,923 (29.7)	15,923 (29.7)	
Types of residence	Living with family	52,899 (94.9)	51,048 (95.4)	103,947 (95.1)	0.000
	Living in relatives’ house	286 (0.5)	242 (0.5)	528 (0.5)	
	Living alone	312 (0.6)	217 (0.4)	529 (0.5)	
	Living in a dormitory	2079 (3.7)	1880 (3.5)	3959 (3.6)	
	Living in a childcare facility	171 (0.3)	147 (0.3)	318 (0.3)	
Age (y)		15 ± 1.8	15.1 ± 1.8	15 ± 1.8	0.192
Height (cm)		165.5 ± 8.6	166.1 ± 8.4	165.8 ± 8.5	0.001
Weight (kg)		58.9 ± 12.9	59.8 ± 13.2	59.3 ± 13.1	0.000
BMI (kg/m^2^)		21.3 ± 3.6	21.5 ± 3.7	21.4 ± 3.6	0.000

**Table 2 ijerph-20-01702-t002:** Participants’ mental state and physical activities.

		2019 N = 55,747 n (%)	2020 N = 53,534 n (%)	Total N = 109,281 n (%)	*p*-Value
Subjective health	Health	39,427 (70.7)	37,702 (70.4)	77,129 (70.6)	0.005
	Moderate	12,367 (22.2)	11,927 (22.3)	24,294 (22.2)	
	Unhealthy	3953 (7.1)	3905 (7.3)	7858 (7.2)	
Perceived stress	Very much	6410 (11.5)	4379 (8.2)	10,789 (9.9)	0.000
	A lot of	15,576 (27.9)	13,666 (25.5)	29,242 (26.8)	
	Little	22,914 (41.1)	23,842 (44.5)	46,756 (42.8)	
	Not really	10,847 (19.4)	11,647 (21.7)	22,494 (20.5)	
Depression experience in the last year	No	40,264 (72.2)	40,138 (75)	80,402 (73.6)	0.000
	Yes	15,483 (27.8)	13,396 (25)	28,879 (26.4)	
High-intensity physical Activity, days/week	NO	17,517 (31.4)	20,394 (38.1)	37,911 (34.7)	0.000
	1–2	19,911 (35.7)	17,872 (33.4)	37,783 (34.6)	
	3–4	10,280 (18.4)	8230 (15.4)	18,510 (17)	
	5≤	8039 (14.4)	7038 (13.1)	15,077 (13.8)	
Muscle -strengthening exercise, days/week	No	28,866 (51.8)	26,810 (50.1)	55,676 (50.9)	0.000
	1–2	14,499 (26)	13,466 (25.1)	27,965 (25.6)	
	3–4	6742 (12.1)	6619 (12.4)	13,361 (12.2)	
	5≤	5640 (10.1)	6639 (12.4)	12,279 (11.2)	
Weight control effort in the last month	Not tried	26,466 (47.5)	24,274 (45.3)	50,740 (46.4)	0.000
	Tried to reduce	18,412 (33)	18,561 (34.7)	36,973 (33.8)	
	Tried to increase	4188 (7.5)	4163 (7.8)	8351 (7.6)	
	Tried to keep	6681 (12)	6536 (12.2)	13,217 (12.1)	
Sleep satisfaction in the last week	Satisfied	3728 (6.7)	5413 (10.1)	91,41 (8.4)	0.000
	Slightly satisfied	8540 (15.3)	10,983 (20.5)	19,523 (17.9)	
	Moderate	18,152 (32.6)	18,195 (34)	36,347 (33.3)	
	Slightly dissatisfied	16,345 (29.3)	13,179 (24.6)	29,524 (27)	
	Dissatisfied	8982 (16.1)	5764 (10.8)	14,746 (13.5)	

**Table 3 ijerph-20-01702-t003:** Participants’ experience of smoking, alcohol consumption, and internet (smartphone) use.

		2019 N = 55,747 n(%)	2020 N = 53,534 n(%)	Total N = 109,281 n(%)	*p*-Value
Smoking	No	48,995 (87.9)	48,104 (89.9)	97,099 (88.9)	0.000
	Yes	6753 (12.1)	5430 (10.1)	12,183 (11.1)	
Liquid e-cigarettes	No	51,961 (93.2)	50,351 (94.1)	102,312 (93.6)	0.000
	Yes	3786 (6.8)	3183 (5.9)	6969 (6.4)	
E-cigarettes	No	53,280 (95.6)	51,857 (96.9)	105,137 (96.2)	0.000
	Yes	2467 (4.4)	1677 (3.1)	4144 (3.8)	
Ease of purchasing cigarettes in the last month	No attempt	52,132 (93.5)	51,065 (95.4)	103,197 (94.4)	0.000
	Not available	1593 (2.9)	1101 (2)	2694 (2.5)	
	Easy	2022 (3.6)	1368 (2.5)	3390 (3.1)	
Secondhand smoke at home, days/week	No	38,007 (68.2)	39,819 (74.4)	77,826 (71.2)	0.000
	1–2	7402 (13.3)	5821 (10.9)	13,233 (12.1)	
	3–4	4403 (7.9)	3230 (6.1)	7633 (7)	
	5–6	1654 (3)	1332 (2.5)	2986 (2.7)	
	Everyday	4282 (7.7)	3332 (6.2)	7614 (7)	
Secondhand smoke at school, days/week	No	44,175 (79.2)	49,802 (93)	93,977 (86)	0.000
	1–2	6526 (11.7)	2281 (4.2)	8807 (8)	
	3–4	2637 (4.8)	714 (1.3)	3351 (3)	
	5–6	892 (1.6)	267 (0.5)	1159 (1.1)	
	Everyday	1517 (2.7)	470 (0.9)	1987 (1.8)	
Secondhand smoke at public place, days/week	NO	26,872 (48.2)	31,255 (58.4)	58,127 (53.2)	0.000
	1–2	15,451 (27.7)	12,637 (23.6)	28,088 (25.7)	
	3–4	8119 (14.6)	5968 (11.1)	14,087 (12.9)	
	5–6	2184 (3.9)	1540 (2.9)	3724 (3.4)	
	Everyday	3121 (5.6)	2134 (4)	5255 (4.8)	
Alcohol consumption	No	34,234 (61.4)	35,744 (66.8)	69,978 (64)	0.000
	Yes	21,514 (38.6)	17,790 (33.2)	39,304 (36)	
Ease of purchasing alcohol in the last month	No attempt	51,012 (91.5)	50,375 (94.1)	101,387 (92.8)	0.000
	Not available	2198 (3.9)	1478 (2.7)	3676 (3.3)	
	Easy	2537 (4.6)	1681 (3.1)	4218 (3.8)	
Weekdays internet (smartphone) usage	No	13,708 (24.6)	1856 (3.5)	15,564 (14.2)	0.000
	Yes	42,040 (75.4)	51,678 (96.5)	93,718 (85.8)	
Weekends internet (smartphone) usage	No	11,930 (21.4)	1684 (3.1)	13,614 (12.5)	0.000
	Yes	43,817 (78.6)	51,850 (96.9)	95,667 (87.5)	

**Table 4 ijerph-20-01702-t004:** Relationship between sleep satisfaction and health status.

	2019	*p*-Value	2020	*p*-Value
	OR (95% CI) *		OR (95% CI) *	
Satisfactory sleep groups	1		1	
Economic changes			1.10 (1.04–1.17)	0.001
Perceived health	2.72 (2.53–2.92)	0.000	2.52 (2.35–2.69)	0.000
Depression experience	2.08 (2.00–2.17)	0.000	2.11 (2.02–2.20)	0.000
High-intensity exercises	0.84 (0.81–0.88)	0.000	0.90 (0.87–0.94)	0.000
Muscle-strengthening exercises	0.84 (0.81–0.87)	0.000	0.89 (0.86–0.93)	0.000
Smoking experience	1.46 (1.40–1.51)	0.000	1.45 (1.40–1.51)	0.000
Alcohol consumption	1.46 (1.38–1.54)	0.000	1.49 (1.40–1.58)	0.000
Secondhand smoke at home	1.15 (1.11–1.20)	0.000	1.19 (1.14–1.24)	0.000
Secondhand smoke at school	1.24 (1.19–1.30)	0.000	1.38 (1.29–1.47)	0.000
Secondhand smoke at public place	1.45 (1.40–1.50)	0.000	1.42 (1.37–1.47)	0.000
Weekdays Internet (smartphone) usage	1.16 (1.11–1.21)	0.000	0.90 (0.82–1.00)	0.044
Weekends internet (smartphone) usage	1.19 (1.14–1.24)	0.000	0.96 (0.87–1.06)	0.432

* Adjusted for age and gender.

## Data Availability

The data provided in this study is accessible to anyone by accessing the website of Korea Disease Control and Prevention Agency [https://www.kdca.go.kr/yhs/] (accessed on 15 October 2022).

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
