# Peer review of "Changes in Sleep Satisfaction of Korean Adolescents in the Pre- and Post-COVID-19 Eras and Its Effects on Health Behaviors"

_ijerph, 2023, doi:10.3390/ijerph20031702_

Round 1

Reviewer 1 Report

Introduction

-Lines 35-36 on page 1: Please consider replacing the comma between “maturity” and “it” by a semicolon or making two separate sentences.

-Lines 38-39 on page 1: The authors mention that “it is critical to determine whether sleep quality , rather than length of sleep, is sufficient to recover from fatigue. Yet, the question they used to measure sleep in their own study measured whether “the amount of sleep in the last seven days was sufficient to recover from fatigue”. In other words, it is a measure of whether sleep duration is sufficient to recover from fatigue. This seems contradictory.

-Line 46 on page 2: Please define OECD.

-Lines 63-64 on page 2: The authors mention that “various factors can affect adolescents’ hours of sleep”, but in the previous sentences of that paragraph they report data on adolescents’ sleep quality not on their sleep duration.

-Lines 69-73 on page 2: I am not sure it is relevant to mention where COVID-19 started and its symptoms. I believe that only mentioning that students in Korea had remote school from March 2020 to October 2021 would be enough.

Materials and Methods

-Lines 93-95 on page 2: Are the two samples (i.e., the one in 2019 and 2020) from the same respondents or are they from new participants who were recruited in 2020? This information is important as it makes more sense to compare differences between 2019 and 2020 if the same participants were asked again the same questions.

-Line 108 on page 3: There is a mistake. Please replace “classifying” by “classified”.

-Lines 124-127 on page 3: Whether sleep duration was sufficient to recover from fatigue was only measured by one item. It would have been much more preferable to use a validated measure of adolescents’ sleep quality like the Adolescent Sleep-Wake Scale or the version for adolescents of the Pittsburgh Sleep Quality Scale. I do not think the item used is comprehensive enough to measure adolescents’ sleep quality. The term “sleep quality” should not be used when referring to the current study.

-Lines 140-142 on page 3: Why did the authors measure Internet use in 2019 and smartphone use in 2020? Is it because they realized that adolescents mainly used their smartphones to access Internet? Both are actually measures of adolescents’ screen time. Smartphones can be used for other purposes than accessing Internet, such as for making phone calls or texting friends or relatives. The authors should find another term to refer to both Internet and smartphone use.

Results

-Table 2 on page 5: The results in Table 1 suggest that there were significant sociodemographic differences between the samples in 2019 and 2020, did the authors adjust for these variables (i.e., add them as covariates) in the analyses reported in Table 2?

-Lines 199-209 and Table 4 on page 8: I am not sure I understand this paragraph and table. Did the authors investigate if the variables listed on the left hand side of the table were associated with satisfactory sleep duration?

Discussion

-Lines 308-310 on page 10: The sentence starting by “Conversely…” appears incomplete.

-Lines 322-325 on page 11: Other serious limitations are that none of the items used to measure health behaviors (physical activity, sleep, screen time, etc.) are validated and sometimes the measures differ between both data collections (e.g., measure of Internet use in 2019 and measure of smartphone use in 2020).

Conclusions

Lines 332-334 on page 11: I do not think the authors can recommend that “at-home exercise programs that help improve quality of sleep can be implemented in schools or community organizations”. Their study does not suggest that increasing physical activity at home will lead to better sleep quality in adolescents and there is currently conflicting data on whether physical activity can improve sleep. Timing (e.g., right before bedtime) and intensity (e.g., high-intensity) would be important factors to consider when looking at whether physical activity can be positively associated with sleep quality. Also, it seems contradictory to recommend at-home exercise programs in schools and community organizations. The schools and community organizations would recommend adolescents to exercise at home instead of at school or in community organizations during future pandemics?

Author Response

Review1

Thank you for your kind feedback. Your advice raised the level of completion of my article.

Introduction

  1. Lines 35-36 on page 1: Please consider replacing the comma between “maturity” and “it” by a semicolon or making two separate sentences.

I revised the sentence as follows.

â–¶Adolescence is an important period for cognitive and psychological development and physical maturity; it is essential to ensure sufficient sleep for a healthy lifestyle.

  1. Lines 38-39 on page 1: The authors mention that “it is critical to determine whether sleep quality rather than length of sleep, is sufficient to recover from fatigue. Yet, the question they used to measure sleep in their own study measured whether “the amount of sleep in the last seven days was sufficient to recover from fatigue”. In other words, it is a measure of whether sleep duration is sufficient to recover from fatigue. This seems contradictory.

The exact meaning of the question was "whether sleep was satisfactory and sufficient to relieve fatigue." Therefore, I modified the sentence as follows.

â–¶the sleep in the last seven days was sufficient to recover from fatigue

  1. Line 46 on page 2: Please define OECD.

I added the following statement to the feedback.

â–¶The Organisation for Economic Co-operation and Development (OECD) is an international organisation that works to build better policies for better lives.

  1. Lines 63-64 on page 2: The authors mention that “various factors can affect adolescents’ hours of sleep”, but in the previous sentences of that paragraph they report data on adolescents’ sleep quality not on their sleep duration.

I modified the sentence as follows.

â–¶ Various factors can affect adolescents’ sleep

  1. Lines 69-73 on page 2: I am not sure it is relevant to mention where COVID-19 started and its symptoms. I believe that only mentioning that students in Korea had remote school from March 2020 to October 2021 would be enough.

â–¶I deleted the sentence according to the feedback.

Coronavirus disease-2019 (COVID-19) originated in Wuhan City, Hubei Province, China, in December 2019 and has since spread worldwide due to its rapid transmission rate. COVID-19 has various respiratory symptoms, including fever, cough, shortness of breath, and pneumonia, and can be asymptomatic or range from mild to severe illness leading to death in extreme cases [23].

Materials and Methods

  1. Lines 93-95 on page 2: Are the two samples (i.e., the one in 2019 and 2020) from the same respondents or are they from new participants who were recruited in 2020? This information is important as it makes more sense to compare differences between 2019 and 2020 if the same participants were asked again the same questions.

2020 respondents are new participants. The survey was conducted by the Korea Centers for Disease Control and Prevention under the Ministry of Health and Welfare, a national institution. This paper was written with free materials provided by the state.

  1. Line 108 on page 3: There is a mistake. Please replace “classifying” by “classified”.

I revised the sentence as follows.

â–¶The types of residence were classified into the five following groups for analyses.

  1. Lines 124-127 on page 3: Whether sleep duration was sufficient to recover from fatigue was only measured by one item. It would have been much more preferable to use a validated measure of adolescents’ sleep quality like the Adolescent Sleep-Wake Scale or the version for adolescents of the Pittsburgh Sleep Quality Scale. I do not think the item used is comprehensive enough to measure adolescents’ sleep quality. The term “sleep quality” should not be used when referring to the current study.

 “Sleep quality” was corrected to “sleep satisfaction”. The title of the paper has been modified accordingly.

  1. Lines 140-142 on page 3: Why did the authors measure Internet use in 2019 and smartphone use in 2020? Is it because they realized that adolescents mainly used their smartphones to access Internet? Both are actually measures of adolescents’ screen time. Smartphones can be used for other purposes than accessing Internet, such as for making phone calls or texting friends or relatives. The authors should find another term to refer to both Internet and smartphone use.

The Youth Health Behavior Survey is an anonymous self-entry online survey conducted on students aged 13 to 18 to understand the health behavior of Korean adolescents. The Korea Centers for Disease Control and Prevention under the Ministry of Health and Welfare has been carrying out it every year since 2005. This study used data from online surveys of youth health behavior in 2019 and 2020. Questions change slightly every year.

Results

  1. Table 2 on page 5: The results in Table 1 suggest that there were significant sociodemographic differences between the samples in 2019 and 2020, did the authors adjust for these variables (i.e., add them as covariates) in the analyses reported in Table 2?

Table 2 shows the survey and analysis in the same form as Table 1. In Tables 1 and 2, cross-analysis was used. The table is separated according to the topic of the question.

  1. Lines 199-209 and Table 4 on page 8: I am not sure I understand this paragraph and table. Did the authors investigate if the variables listed on the left hand side of the table were associated with satisfactory sleep duration?

Table 4 compares variables based on groups with satisfactory sleep and those without sleep. The reason why I chose that variable is because it's the subject of this paper. It is also a selection from the questionnaires already conducted by national institutions.

Discussion

  1. Lines 308-310 on page 10: The sentence starting by “Conversely…” appears incomplete.

Conversely, if a person consumes fewer sugary drinks, the virtuous cycle may be repeated because more sleep.

â–¶Conversely, if a person consumes fewer sugary drinks, the virtuous cycle may be repeated because of more sleep.

  1. Lines 322-325 on page 11: Other serious limitations are that none of the items used to measure health behaviors (physical activity, sleep, screen time, etc.) are validated and sometimes the measures differ between both data collections (e.g., measure of Internet use in 2019 and measure of smartphone use in 2020).

I added it to the limitation part as follows:

â–¶The limitation of this study is that the first self-report survey can lead to recall errors. The second cross-sectional study is that there is a limit to explaining the causal relationship between health behavior and sleep. Lastly, since it is a study conducted with data already collected, it is difficult to know the impact on individual sleep. In addition, as a survey conducted by a national institution, the questionnaire was written using the Internet in 2019 and the questionnaire was written using a smartphone in 2020.

Conclusions

  1. Lines 332-334 on page 11: I do not think the authors can recommend that “at-home exercise programs that help improve quality of sleep can be implemented in schools or community organizations”. Their study does not suggest that increasing physical activity at home will lead to better sleep quality in adolescents and there is currently conflicting data on whether physical activity can improve sleep. Timing (e.g., right before bedtime) and intensity (e.g., high-intensity) would be important factors to consider when looking at whether physical activity can be positively associated with sleep quality. Also, it seems contradictory to recommend at-home exercise programs in schools and community organizations. The schools and community organizations would recommend adolescents to exercise at home instead of at school or in community organizations during future pandemics?

I revised the sentence as follows.

â–¶Since there may be environmental changes such as the COVID-19 situation in the future, schools or local organizations should prepare for such situations and exercise programs.

Reviewer 2 Report

It is good quality online based survey study. The big advantage is high number of study participants. However I found the following major flaws:

1. Please use the following title: Changes in health behaviors of Korean adolescents in the pre-and post-COVID-19 eras and its effects on sleep quality

2. Please add term "COVID-19" to the keywords.

3. Please remove the following sentences from Introduction: "Coronavirus disease-2019 (COVID-19) originated in Wuhan City, Hubei Province, China, in December 2019 and has since spread worldwide due to its rapid transmission rate. COVID-19 has various respiratory symptoms, including fever, cough, shortness of breath, and pneumonia, and can be asymptomatic or range from mild to severe illness leading to death in extreme cases [23]." It is well known facts. Therefore it is unnecessary.

4. Authors wrote "Several studies have reported on the importance of adequate sleep quantity and sleep quality." but Authors did not cite the studies and Authors should do it.

5. I recommend to add a paragraph about latest and reliable studies related to the topic including risk factors and groups of COVID-19, relationship of COVID-19 with general health, mental health and sleep and then write why Authors study is new and important. I suggest the following articles:

Martynowicz H, Jodkowska A, PorÄ™ba R, Mazur G, WiÄ™ckiewicz M. Demographic, clinical, laboratory, and genetic risk factors associated with COVID-19 severity in adults: A narrative review. Dent Med Probl. 2021;58(1):115–121. doi:10.17219/dmp/131795

StraburzyÅ„ski M, Kuca-Warnawin E, Waliszewska-ProsóÅ‚ M. COVID-19-related headache and innate immune response - a narrative review [published online ahead of print, 2022 Jun 27]. Neurol Neurochir Pol. 2022;10.5603/PJNNS.a2022.0049.

Wieckiewicz, M., Danel, D., Pondel, M. et al. Identification of risk groups for mental disorders, headache and oral behaviors in adults during the COVID-19 pandemic. Sci Rep 11, 10964 (2021). https://doi.org/10.1038/s41598-021-90566-z

StraburzyÅ„ski M, Nowaczewska M, Budrewicz S, Waliszewska-ProsóÅ‚ M. COVID-19-related headache and sinonasal inflammation: A longitudinal study analysing the role of acute rhinosinusitis and ICHD-3 classification difficulties in SARS-CoV-2 infection. Cephalalgia. 2022;42(3):218-228.

Kolcakoglu K, Yucel G. Anxiety and harmful oral habits in preschool children during the 2020 first-wave COVID-19 lockdown in Turkey. Dent Med Probl. 2021;58(4):433–439. doi:10.17219/dmp/142284

6.  Authors have to write „Based on an earlier study which reported that sleep loss was compensated by increasing the total hours of sleep in case of low sleep efficiency [17], this study aimed to analyze the sleep quality and to investigate the effect of the COVID-19 situation on Korean adolescents' sleep quality and whether such effects were related to adolescents' psychological state, physical activity, smoking, alcohol consumption, and internet (smartphone) use.”

7. Title, aim of the study and conclusions have to correspond one to each other.

8. Authors have to precisely describe the online survey distribution (software, online tools etc). Authors have to provide a source of mailing list of potential study participants. Authors have to describe inclusion and exclusion criteria for study participants.

9. Authors have to calculate sample power and size and odds ratio for tested comparisons to show strength of tested relationships.

Author Response

Review2

Thank you for your detailed feedback. My thesis has been completed more with your guidance.

  1. Please use the following title: Changes in health behaviors of Korean adolescents in the pre-and post-COVID-19 eras and its effects on sleep quality.

I modified it according to the feedback.

  1. Please add term "COVID-19" to the keywords.

I added a keyword according to the feedback.

  1. Please remove the following sentences from Introduction: "Coronavirus disease-2019 (COVID-19) originated in Wuhan City, Hubei Province, China, in December 2019 and has since spread worldwide due to its rapid transmission rate. COVID-19 has various respiratory symptoms, including fever, cough, shortness of breath, and pneumonia, and can be asymptomatic or range from mild to severe illness leading to death in extreme cases [23]." It is well known facts. Therefore it is unnecessary.

â–¶I deleted the sentence according to the feedback.

Coronavirus disease-2019 (COVID-19) originated in Wuhan City, Hubei Province, China, in December 2019 and has since spread worldwide due to its rapid transmission rate. COVID-19 has various respiratory symptoms, including fever, cough, shortness of breath, and pneumonia, and can be asymptomatic or range from mild to severe illness leading to death in extreme cases [23].

  1. Authors wrote "Several studies have reported on the importance of adequate sleep quantity and sleep quality." but Authors did not cite the studies and Authors should do it.

This sentence means that there has been such research until now. It's about research trends. To prevent confusion, I will present a reference.

  1. I recommend to add a paragraph about latest and reliable studies related to the topic including risk factors and groups of COVID-19, relationship of COVID-19 with general health, mental health and sleep and then write why Authors study is new and important. I suggest the following articles:

Following the feedback, I added the following sentence.

▶COVID-19 causes headaches caused by acute sinusitis along with respiratory diseases [Straburzynski, 2021]. COVID-19 and vaccines can cause immune response symptoms [Straburzyński, 2022].

In particular, it affects the physical activity and psychological health of adolescents [10].

The COVID-19 Pandemic period increases screen time and affects eating habits, physical activity, and sleep quality [23].

It is important to compare sleep conditions before and after COVID-19 and analyze health behavior.

  1. Authors have to write “Based on an earlier study which reported that sleep loss was compensated by increasing the total hours of sleep in case of low sleep efficiency [17], this study aimed to analyze the sleep quality and to investigate the effect of the COVID-19 situation on Korean adolescents' sleep quality and whether such effects were related to adolescents' psychological state, physical activity, smoking, alcohol consumption, and internet (smartphone) use.”

“sleep quality” was corrected to “sleep satisfaction”.

â–¶Based on an earlier study which reported that sleep loss was compensated by increasing the total hours of sleep in case of low sleep efficiency [17], this study aimed to analyze the sleep satisfaction and to investigate the effect of the COVID-19 situation on adolescents' sleep satisfaction and whether such effects were related to adolescents' psychological state, physical activity, smoking, alcohol consumption, and internet (smartphone) use.

  1. Title, aim of the study and conclusions have to correspond one to each other.

â–¶I modified title according to the feedback.

Title : Changes in sleep satisfaction of Korean adolescents in the pre-and post-COVID-19 eras and its effects on health behaviors

Aim of the study : This study aimed to analyze the sleep satisfaction and to investigate the effect of the COVID-19 situation on adolescents' sleep satisfaction and whether such effects were related to adolescents' psychological state, physical activity, smoking, alcohol consumption, and internet (smartphone) use.

Conclusions : During the COVID-19 pandemic, non-face-to-face classes were conducted, and adolescents' degree of post-sleep fatigue relief and sleep quality improved. Their psychological state improved, the amount of physical activity increased mainly for muscle-strengthening exercises, and the average body weight increased. Smoking, secondhand smoke exposure, and alcohol consumption decreased, while internet (smartphone) usage increased.

  1. Authors have to precisely describe the online survey distribution (software, online tools etc). Authors have to provide a source of mailing list of potential study participants. Authors have to describe inclusion and exclusion criteria for study participants.

The survey was conducted by a state agency. It is conducted by the Korea Centers for Disease Control and Prevention under the Ministry of Health and Welfare, the agency that conducted the survey. This content has been added to the research method section.

â–¶The Youth Health Behavior Survey is an anonymous self-entry online survey conducted on students aged 13 to 18 to understand the health behavior of Korean adolescents. The Korea Centers for Disease Control and Prevention under the Ministry of Health and Welfare has been carrying out it every year since 2005. This study used data from online surveys of youth health behavior in 2019 and 2020.

  1. Authors have to calculate sample power and size and odds ratio for tested comparisons to show strength of tested relationships.

In this paper, 2019 survey participants and 2020 participants are different. This paper is made up of data from the survey that has already been conducted. The odds ratios are specified in Table 4.

2019

p-value

2020

p-value

OR(95% CI)*

OR(95% CI)*

Satisfactory sleep groups

1

1

Economic changes

1.10(1.04 – 1.17)

0.001

Perceived health

2.72(2.53 – 2.92)

0.000

2.52(2.35 – 2.69)

0.000

Depression experience

2.08(2.00 – 2.17)

0.000

2.11(2.02 – 2.20)

0.000

High-intensity exercises

0.84(0.81 – 0.88)

0.000

0.90(0.87 – 0.94)

0.000

Muscle-strengthening exercises

0.84(0.81 – 0.87)

0.000

0.89(0.86 – 0.93)

0.000

Smoking experience

1.46(1.40 – 1.51)

0.000

1.45(1.40 – 1.51)

0.000

Alcohol consumption

1.46(1.38 – 1.54)

0.000

1.49(1.40 – 1.58)

0.000

Secondhand smoke at home

1.15(1.11 – 1.20)

0.000

1.19(1.14 – 1.24)

0.000

Secondhand smoke at school

1.24(1.19 – 1.30)

0.000

1.38(1.29 – 1.47)

0.000

Secondhand smoke at public

place

1.45(1.40 – 1.50)

0.000

1.42(1.37 – 1.47)

0.000

Weekdays

Internet (smartphone) usage

1.16(1.11 – 1.21)

0.000

0.90(0.82 – 1.00)

0.044

Weekends internet (smartphone) usage

1.19(1.14 – 1.24)

0.000

0.96(0.87 – 1.06)

0.432

Round 2

Reviewer 1 Report

The authors did a great job at integrating most of my previous comments and the manuscript has improved in terms of clarity. I still have a few comments.

Abstract

Last sentence of the abstract: I am not sure I agree with this sentence. The authors’ study showed an association between sleep satisfaction and physical activity, but it does not mean that at-home exercise programs will result in improved sleep satisfaction in adolescents in future pandemic situations.

Introduction

Line 48 on page 2: It is more common to define an abbreviation (like OECD) upon its first use in the text, not in the following sentence.

Lines 84-86 on page 2: The sentence starting with “The COVID-19 Pandemic period” is written in the present tense, while the rest of the studies on COVID-19 of that paragraph (i.e., the sentences after) are written in the past tense. Is this on purpose? Do the authors mean that the COVID-19 pandemic is still affecting people’s screen time, eating habits, physical activity, and sleep quality? If not, it should be in the past tense like the other sentences of that paragraph.

Lines 93-94 on page 2: The last sentence of that paragraph seems to come out of nowhere. Maybe add a “thus” or “therefore” (e.g., It is therefore important to compare sleep conditions…).

Methods

Line 170 on page 4: For consistency with the rest of the text, please remove “hours” (i.e., For comparison of health status according to satisfactory sleep among adolescents…).

Results

Lines 188-189 on page 4: The authors report the average height and weight of adolescents. It might be more informative to report the average body mass index (BMI) in kg/m squared. This would allow readers to have a general idea if a high percentage of the sample is underweight, healthy weight, overweight and/or obese. Same thing for the results reported in Table 1. This information is important as weight can have an impact on sleep. People with overweight and obesity tend to have worse sleep quality compared to those with healthy weight. Overweight and obesity are associated with risks of sleep apnea. People with undiagnosed sleep apnea would most likely be dissatisfied with their sleep. BMI could be added as a covariate in the analyses.

Discussion

Lines 286-288 on page 13: The authors mention that the average weight increased by 1 kg, but does this really warrant the necessity of efforts to decrease body weight? Again, it would be more informative to have information on the percentage of adolescents with overweight/obesity. If a 1 kg increase in body weight had no impact on the percentage of adolescents with overweight/obesity, then it does not really warrant public health efforts to decrease body weight. It might just be the result of normal growth/maturation.

Lines 348-349: I think there is a word missing (i.e., Conversely, if a person consumes fewer sugary drinks, the virtuous cycle may be repeated because OF more sleep).

Lines 378-380: I agree that schools and local organizations should prepare for future pandemics, but I am not sure I understand the part about exercise programs. Do the authors mean that schools and local organizations should prepare exercise programs in case of future pandemics? If so, please explain the rationale behind this.

Author Response

Dear reviewer,

Thank you for your kind feedback.
I revised the paper again.

Last sentence of the abstract: I am not sure I agree with this sentence. The authors’ study showed an association between sleep satisfaction and physical activity, but it does not mean that at-home exercise programs will result in improved sleep satisfaction in adolescents in future pandemic situations.

â–¶I deleted the sentence.

At-home exercise programs that improve sleep satisfaction can be implemented in schools or community organizations, even in future pandemic situations.

Introduction

Line 48 on page 2: It is more common to define an abbreviation (like OECD) upon its first use in the text, not in the following sentence.

â–¶I modified the sentence according to the feedback.

A survey on subjective happiness conducted among adolescents in major The Organisation for Economic Co-operation and Development (OECD) countries reported that Korean adolescents had low happiness levels and were under high stress due to excessive competition for university entrance exams [7]. OECD is an international organisation that works to build better policies for better lives.

Lines 84-86 on page 2: The sentence starting with “The COVID-19 Pandemic period” is written in the present tense, while the rest of the studies on COVID-19 of that paragraph (i.e., the sentences after) are written in the past tense. Is this on purpose? Do the authors mean that the COVID-19 pandemic is still affecting people’s screen time, eating habits, physical activity, and sleep quality? If not, it should be in the past tense like the other sentences of that paragraph.

â–¶Currently, there is no non-face-to-face class, so the sentence had to be written in the past tense. Therefore, I modified it.

The COVID-19 Pandemic period increased screen time and affected eating habits, physical activity, and sleep quality [25].

Lines 93-94 on page 2: The last sentence of that paragraph seems to come out of nowhere. Maybe add a “thus” or “therefore” (e.g., It is therefore important to compare sleep conditions…).

â–¶I revised the sentence according to the feedback.

It is therefore important to compare sleep conditions before and after COVID-19 and analyze health behavior.

Methods

Line 170 on page 4: For consistency with the rest of the text, please remove “hours” (i.e., For comparison of health status according to satisfactory sleep among adolescents…).

â–¶I deleted the word according to the correction.

For comparison of health status according to the satisfactory sleep hours among adolescents in the pre- and post- COVID-19 eras,…

Results

Lines 188-189 on page 4: The authors report the average height and weight of adolescents. It might be more informative to report the average body mass index (BMI) in kg/m squared. This would allow readers to have a general idea if a high percentage of the sample is underweight, healthy weight, overweight and/or obese. Same thing for the results reported in Table 1. This information is important as weight can have an impact on sleep. People with overweight and obesity tend to have worse sleep quality compared to those with healthy weight. Overweight and obesity are associated with risks of sleep apnea. People with undiagnosed sleep apnea would most likely be dissatisfied with their sleep. BMI could be added as a covariate in the analyses.

â–¶I revised the table according to the feedback. I also added the following sentence.

The average age, height, weight, and BMI were 15 years, 165.8 cm, 59.3 kg, and 21.4kg/m2 respectively.

Discussion

Lines 286-288 on page 13: The authors mention that the average weight increased by 1 kg, but does this really warrant the necessity of efforts to decrease body weight? Again, it would be more informative to have information on the percentage of adolescents with overweight/obesity. If a 1 kg increase in body weight had no impact on the percentage of adolescents with overweight/obesity, then it does not really warrant public health efforts to decrease body weight. It might just be the result of normal growth/maturation.

â–¶A person's weight did not increase by 1kg in a year, but the average weight in a group of the same age increased year-on-year. This is not the result of growth. The average weight of the group in 2020 is higher than that of the group in 2019. I also came to that conclusion because not only did I increase my weight average, but also decreased the number of days of high-intensity physical activity.

Lines 348-349: I think there is a word missing (i.e., Conversely, if a person consumes fewer sugary drinks, the virtuous cycle may be repeated because OF more sleep).

â–¶I added words according to the feedback.

Conversely, if a person consumes fewer sugary drinks, the virtuous cycle may be repeated because of more sleep [46].

Lines 378-380: I agree that schools and local organizations should prepare for future pandemics, but I am not sure I understand the part about exercise programs. Do the authors mean that schools and local organizations should prepare exercise programs in case of future pandemics? If so, please explain the rationale behind this.

â–¶I modified the sentence as follows according to the feedback.

Since there may be environmental changes such as the COVID-19 situation in the future, schools or local organizations should prepare for such situations and exercise programs.

Reviewer 2 Report

The manuscript has been significantly improved. I don't have further comments.

Author Response

Thank you for your kind feedback.